# MRRL: Modifying the Reference via Reinforcement Learning for Non-Autoregressive Joint Multiple Intent Detection and Slot Filling

**Xuxin Cheng[†], Zhihong Zhu[†], Bowen Cao,**
**Qichen Ye, Yuexian Zou[*]**
School of ECE, Peking University, China
{chengxx, zhihongzhu, cbw2021}@stu.pku.edu.cn
{yeeeqichen, zouyx}@pku.edu.cn

## Abstract

With the rise of non-autoregressive approach, some non-autoregressive models for joint multiple intent detection and slot filling have obtained the promising inference speed. However, most existing SLU models (1) suffer from the multi-modality problem that leads to reference intents and slots may not be suitable for training; (2) lack of alignment between the correct predictions of the two tasks, which extremely limits the overall accuracy. Therefore, in this paper, we propose **M**odifying the **R**eference via **R**einforcement **L**earning (MRRL), a novel method for multiple intent detection and slot filling, which introduces a modifier module and employs reinforcement learning. Specifically, we try to provide the better training target for the non-autoregressive SLU model via modifying the reference based on the output of the non-autoregressive SLU model, and propose a suitability reward to ensure that the output of the modifier module could fit well with the output of the non-autoregressive SLU model and does not deviate too far from the reference. In addition, we also propose a compromise reward to realize a flexible trade-off between the two subtasks. Experiments on two multi-intent datasets and non-autoregressive baselines demonstrate that our MRRL could consistently improve the performance of baselines. More encouragingly, our best variant achieves new state-of-the-art results, outperforming the previous best approach by 3.6 overall accuracy on MixATIS dataset.

## 1 Introduction

As a crucial task in dialogue systems, spoken language understanding (SLU) aims to understand the user's current goal through constructing semantic frames (Tur and De Mori, 2011; Young et al., 2013; Zhu et al., 2023b; He and Garner, 2023b). Intent detection and slot filling are two common subtasks of SLU, where intent detection is an utterance-level classification task and slot filling could be regarded as a sequence labeling task (He and Garner, 2023a; Cheng et al., 2023a).

However, an utterance often contains more than a just single intent in the real scenarios (Zhu et al., 2023a; Xing and Tsang, 2023; Cheng et al., 2023f). With this in mind, multi-intent SLU has received more and more attention (Xu and Sarikaya, 2013; Kim et al., 2017; Shet et al., 2019). Gangadharaiah and Narayanaswamy (2019) proposes a multi-task framework to jointly model intent detection and slot filling. Qin et al. (2020) introduces graph attention networks (GAT) (Velickovic et al., 2018) to develop a fine-grained multi-intent prediction framework called AGIF, which aims to incorporate intent information into the slot filling decoding process in an adaptive manner. Qin et al. (2021b) proposes a novel global-locally graph interaction network GL-GIN, which appplies the non-autoregressive modeling techniques to parallelize the decoding process, resulting in significant speedup compared to the traditional autoregressive SLU models. Xing and Tsang (2022a) further designs a two-stage SLU framework and achieves the mutual guidance between intent and slot, which enhances the overall accuracy of SLU. Song et al. (2022) explores attention mechanisms to extract the relevant information from the independent utterance contexts and capture shared label-specific features across all utterances in the training set. Xing and Tsang (2022b) proposes ReLa-Net, which utilizes a heterogeneous label graph to represent the statistical dependencies and hierarchies. Cheng et al. (2023d) applies contrastive learning to explore and leverage the inherent relationships in multi-intent SLU. Cheng et al. (2023a) proposes a scope-sensitive SLU model SS-RAN to reduce the distraction of the out-of-scope tokens and mitigate the error propagation problem caused by the bidirectional interaction.

Though existing non-autoregressive multi-intent SLU models have made the promising progress, we

---

[†] Equal contribution.
[*] Corresponding author.

| Tokens | Possibility | Reference |
|--------|-------------|-----------|
| new | `intent:atis_quantity, atis_flight`
`slot:    B-fromloc.city_name` | atis_distance, atis_day_name
B-city_name |
| guardia | `intent:atis_airport, atis_airline`
`slot:   B-fromloc.airport_name` | atis_distance, atis_day_name
I-airport_name |
| downtown | `intent:  atis_city, atis_airfare`
`slot:       B-city_name` | atis_distance, atis_day_name
O |

Table 1: The examples of the multi-modality problem. The gold intent label of the utterance is utilized as the intent of each token in the utterance.

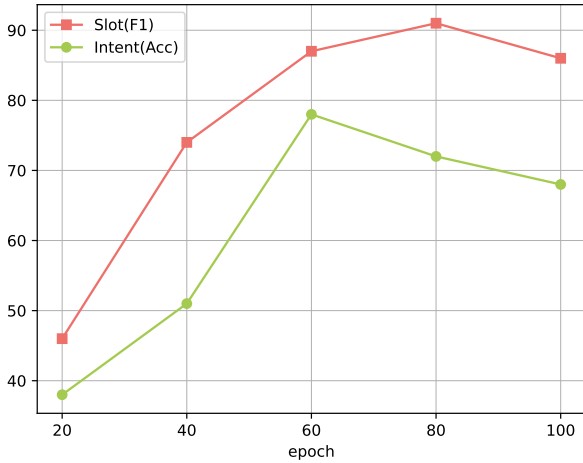

Figure 1: An example of the misalignment between the correct predictions of intent detction and slot filling. As the F1 score of slot filling (red) increases, the accuracy of intent detection (green) may decrease, which limits the overall accuracy of the utterance.

find that most of them still face two issues:

(1) **Suffer from the multi-modality problem.** Although non-autoregressive models have proven the effectiveness in terms of high inference speed, they still suffer from the multi-modality problem, which has been pointed out in several prior works in other tasks (Ran et al., 2020; Zhang et al., 2022a). However, this serious problem is still ignored in non-autoregressive SLU task. As shown in Table 1, for each token, there may be multiple possible correct slots. Besides, due to the widespread use of the token-level intent detection decoder, this problem also occurs in intent detection. Despite the utilization of GAT (Velickovic et al., 2018), these models still have little prior knowledge about the reference during the inference progress, which leads to some errors not commonly seen in autoregressive SLU models. For example, an I- slot might erroneously appear before its corresponding B- slot in the slot sequence output by a non-autoregressive model. As a result, the original reference intents and slots are not very suitable for model training.

(2) **Lack of the alignment between the correct predictions of the two subtasks.** Most of existing SLU models decode the hidden stats of the two subtasks independently without leveraging the correlations between them, which leads to the misalignment of the correct predictions of the two subtasks. As shown in Figure 1, the F1 score of slot filling and the accuracy of intent detection might increase or decrease asynchronously. Overall accuracy is an important metric and it denotes the ratio of the utterances for which both intents and slots are predicted correctly. Due to the lack of alignment, overall accuracy on utterance-level semantic frame parsing is much worse than these two subtasks, which is not conducive to deploying the SLU model in actual scenarios.

In this paper, we propose a novel model termed MRRL to tackle the above two issues, which introduces a modifier and applies reinforcement learn-

ing to provide a better training target for the non-autoregressive model. Since the reference intents and slots may not be suitable for training, we modify the reference intents and slots according to the output of the non-autoregressive SLU model. For the first problem, we propose a suitability reward to ensure that the output of the modifier fits well with the output of the non-autoregressive model and still maintain the original information. The first part of suitability reward is related to the training loss on the output of the modifier, and the second part is the similarity between the reference and the output of the modifier. For the second problem, we propose a compromise reward to improve the overall accuracy. We directly utilize the overall accuracy as the compromise reward to achieve a flexible trade-off between the accuracy of intent detection and the F1 score of slot filling. Experiment results show that MRRL consistently improves the performance of baselines on two benchmark datasets MixATIS and MixSNIPS (Hemphill et al., 1990; Coucke et al., 2018; Qin et al., 2020). Further analysis also verifies the advantages of our method.

The contributions of our work are three-fold:

- To the best of our knowledge, our work is the first attempt to employ reinforcement learning to solve the multi-modality problem and the misalignment problem in non-autoregressive multi-intent SLU models.

- We propose a novel method termed MRRL for non-autoregressive multi-intent SLU, which introduces a modifier and applies reinforce-

ment learning to modify the reference utterance into a form suitable for model training.

- Experimental results demonstrate that MRRL can efficiently improve the performance of the baselines and the best variant achieves the new state-of-the-art results.

## 2 Background

In this section, we first introduce the problem definition of the multi-intent SLU task and then introduce several multi-intent SLU models.

Given an input utterance $\boldsymbol{x} = (x_1, x_2, \ldots, x_n)$, where $n$ denotes the length of the utterance $\boldsymbol{x}$, multiple intent detection is a multi-label classification task and the token-level predicted intent sequence is denoted as $\boldsymbol{y}^{I'} = (y^{(1,I')}, y^{(2,I')}, \ldots, y^{(n,I')})$. The final utterance-level predicted intent sequence $\boldsymbol{y}^I$ is obtained by the token-level intent voting strategy (Qin et al., 2021b; Xing and Tsang, 2022a). The reference intent sequence is denoted as $\hat{\boldsymbol{y}}^I = (\hat{y}^{(1,I)}, \hat{y}^{(2,I)}, \ldots, \hat{y}^{(m,I)})$, where $m$ denotes the number of intents in $\boldsymbol{x}$. Slot filling is a sequence labeling task (Qin et al., 2022; Cheng et al., 2023c; Zhu et al., 2023c). The predicted slot sequence is denoted as $\boldsymbol{y}^S = (y^{(1,S)}, y^{(2,S)}, \ldots, y^{(n,S)})$ and the reference slot sequence is denoted as $\hat{\boldsymbol{y}}^S = (\hat{y}^{(1,S)}, \hat{y}^{(2,S)}, \ldots, \hat{y}^{(n,S)})$. For simplicity, we use $\hat{\boldsymbol{y}}$ to denote the union of $\hat{\boldsymbol{y}}^I$ and $\hat{\boldsymbol{y}}^S$.

### 2.1 Multi-Intent Spoken Language Understanding

Due to the interaction between the two subtasks of multiple intent detection and slot filling, joint models are widely used to consider the two tasks and update parameters. The multiple intent detection objective is defined as:

$$\text{CE}(\hat{y}, y) = \hat{y}\log(y) + (1 - \hat{y})\log(1 - y) \quad (1)$$

$$\mathcal{L}_I = -\sum_{i=1}^{n}\sum_{j=1}^{N_I}\text{CE}(\hat{y}_i^{(j,I)}, y_i^{(j,I)}) \quad (2)$$

where $N_I$ denotes the number of single intent labels, $\hat{y}_i^{(j,I)}$ denotes the reference intent, and $y_i^{(j,I)}$ denotes its corresponding predicted intent.

Similarly, the slot filling objective is defined as:

$$\mathcal{L}_S = -\sum_{i=1}^{n}\sum_{j=1}^{N_S}\hat{y}_i^{(j,S)}\log\left(y_i^{(j,S)}\right) \quad (3)$$

where $N_S$ denotes the number of slot labels, $\hat{y}_i^{(j,S)}$ denotes the reference slot, and $y_i^{(j,S)}$ denotes its corresponding predicted slot.

The final joint objective is formulated as:

$$\mathcal{L} = \alpha\mathcal{L}_I + \beta\mathcal{L}_S \quad (4)$$

where $\alpha$ and $\beta$ are hyper-parameters.

### 2.2 AGIF

AGIF (Qin et al., 2020) is a token-level adaptive interaction network that implements fine-grained integration of multi-intent information. An intent-slot graph interaction layer is used to model the strong correlation between slots and intents. Such an interaction layer is applied adaptively to each token of an utterance, with the advantage that relevant intent information can be automatically extracted. Restricted by the autoregressive paradigm, during the inference, the previously predicted tokens must be fed to the decoder to generate the next token step by step, which leads to slower inference speed.

### 2.3 GL-GIN

GL-GIN (Qin et al., 2021b) is a global-locally graph-interaction network, including a local slot-aware graph layer and a global intent-slot interaction layer. Owing to the non-autoregressive architecture, GL-GIN achieves to generate intents and slots sequence simultaneously, thus increasing the inference speed. However, we find that it is not enough to just rely on the local graph interaction layer to model the slot dependencies, which limits the performance (see Sec.4.5 for more details).

### 2.4 Co-guiding Net

Co-guiding Net (Xing and Tsang, 2022a) is a two-stage framework that allows intent detection and slot filling to learn from each other. The first stage produces initial estimated labels for the two tasks and the second stage leverages estimated labels as prior label information. Two heterogeneous graph attention networks are proposed to work on the two aforementioned graphs for modeling the guidance between intent and slot.

### 2.5 ReLa-Net

ReLa-Net (Xing and Tsang, 2022b) improves joint multiple intent detection and slot filling from a new perspective, which exploits the label typologies and relations through a heterogeneous label graph and a recurrent heterogeneous label matching network. The heterogeneous label graph includes both the global statistical dependencies and slot label hierarchies, which is proposed to represent the statistical

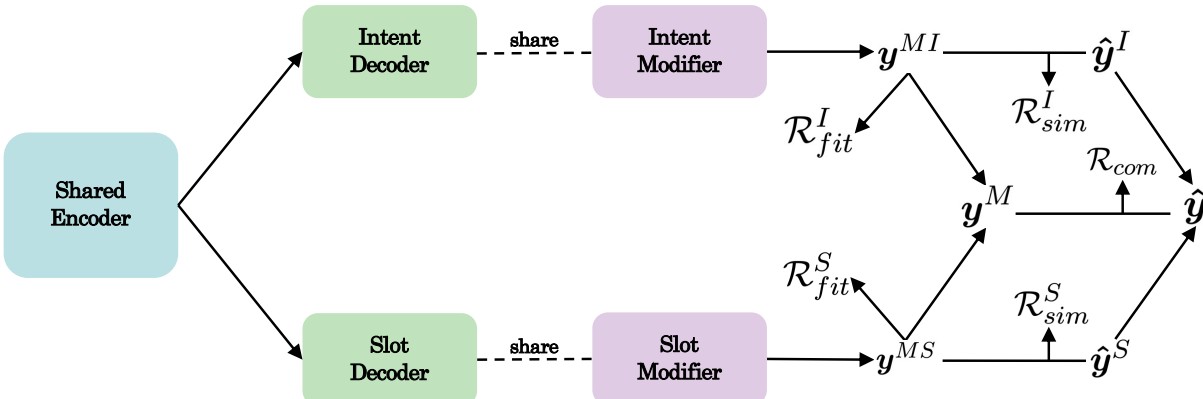

Figure 2: The main architecture of MRRL. Two modifiers are trained to optimize the suitability reward and the compromise reward, where suitability reward is the weighted sum of fit reward and similarity reward. Fit reward relates to the joint training loss of the modifiers and similarity reward relates to the similarity with the reference. Compromise reward is related to the overall accuracy.

dependencies and hierarchies in rich relations. And a recurrent heterogeneous label matching network is proposed to end-to-end capture the beneficial information from the heterogeneous label graph and use them for tackling the joint task.

## 3 Approach

In this section, we first present the overview of our model (§3.1), which introduces the modifier module including a intent modifier and a slot modifier into the original architecture. Then, we introduce the pretrain-finetune paradigm (§3.2), where pretraining could establish a better initial state for the fine-tuning of the modifier module. Finally, we introduce the reinforcement learning module, which is applied to make the output of the modifier more suitable for model training (§3.3). Figure 2 illustrates the overview of our proposed method.

### 3.1 Model Overview

To alleviate the problem of multi-modality and misalignment, in addition to the conventional encoder-decoder architecture, we introduce a modifier module including a intent modifier and a slot modifier to modify the reference according to the output of the non-autoregressive model. Note that the modifier module only affect the training stage, so the inference cost does not change.

The modifier module is designed as a key component capable of transforming the reference into a more suitable form for training. The transformation process takes into account both the reference itself and the output generated by the non-autoregressive model. Through leveraging this dual information,

the modifier module has the potential to optimize the reference, making it more aligned with the training objectives and enhancing the overall training process. Thus, we utilize two non-autoregressive decoders as the architecture of the modifier module. Similarly, the token-level predicted intent sequence of the intent modifier is denoted as $\boldsymbol{y}^{MI'} = (y^{(1,MI')}, y^{(2,MI')}, \ldots, y^{(n,MI')})$, the output of the intent modifier $\boldsymbol{y}^{MI}$ is also obtained by the voting technique. The output of the slot modifier is denoted as $\boldsymbol{y}^{MS} = (y^{(1,MS)}, y^{(2,MS)}, \ldots, y^{(n,MS)})$, and the union of $\boldsymbol{y}^{MI}$ and $\boldsymbol{y}^{MS}$ is denoted as $\boldsymbol{y}^{M}$. We use $P_M$ to denote the probability distribution of the modifier module:

$$P_M(\boldsymbol{y}^{MI'}, \boldsymbol{y}^{MS}|\boldsymbol{x}, \hat{\boldsymbol{y}}) = \prod_{i=1}^{n} p_M(y^{(i,MI')},$$
$$y^{(i,MS)}|\boldsymbol{x}, \hat{\boldsymbol{y}}) \quad (5)$$

Diverging from conventional SLU models, the non-autoregressive SLU model is trained with the output of the modifier module $\boldsymbol{y}^{MI}$ and $\boldsymbol{y}^{MS}$ rather than the original reference $\hat{\boldsymbol{y}}^I$ and $\hat{\boldsymbol{y}}^S$. Then the multiple intent detection objective $\mathcal{L}_{MI}$ and slot filling objective $\mathcal{L}_{MS}$ are formulated as:

$$\mathcal{L}_{MI} = -\sum_{i=1}^{n}\sum_{j=1}^{N_I} \text{CE}(\hat{y}_i^{(j,I)}, y_i^{(j,MI)}) \quad (6)$$

$$\mathcal{L}_{MS} = -\sum_{i=1}^{n}\sum_{j=1}^{N_S} \hat{y}_i^{(j,S)} \log\left(y_i^{(j,MS)}\right) \quad (7)$$

where $y_i^{(j,MI)}$ denotes the predicted intent of the intent modifier and $y_i^{(j,MS)}$ denotes the predicted slot

of the slot modifier. Then the final joint objective of the modifier module $\mathcal{L}_M$ is defined as:

$$\mathcal{L}_M = \alpha \mathcal{L}_{MI} + \beta \mathcal{L}_{MS} \tag{8}$$

### 3.2 Pretrain-Finetune Paradigm

When directly applying reinforcement learning to train the modifier, a common problem arises where the modifier become trapped in a non-optimal state, resulting in the generation of sequences with meaningless intents and slots. The issue is inherent to reinforcement learning due to its sensitivity to initial states (Mihatsch and Neuneier, 2002; Fei et al., 2020; Shao et al., 2023).

To address this challenge, we utilize a pretrain-finetune paradigm. During the pre-training phase, we employ a joint objective $\mathcal{L}$ as defined in Eq.4 to establish a better initial state for the modifier. This pre-training strategy aims to provide the modifier with a more favorable starting point, making subsequent fine-tuning easier and reducing the likelihood of falling into local optimization.

Following the pre-training phase, we proceed to fine-tune the modifier using reinforcement learning, incorporating suitability reward and compromise reward to further guide the learning process. This fine-tuning approach enables the designed modifier to refine its behavior and adjust its outputs based on the reinforcement signals received.

### 3.3 Reinforcement Learning

The primary objective of the modifier module is to generate a training target which is better suited for non-autoregressive models. Motivated by Wu et al. (2018); Rao et al. (2021); Lu et al. (2022); Shao et al. (2023), we quantify the requirements for the modifier module into two reward functions and optimize them via reinforcement learning. The level of appropriateness of this target could be quantified using two reward functions.

Firstly, it is crucial for the output of the modifier module to align closely with the output of the SLU model. We employ a fitting reward $\mathcal{R}_{fit}$ that shares a formal resemblance to the loss function $\mathcal{L}_M$, aiming to incentivize the reduction of the training loss.

For the output of the intent modifier $\boldsymbol{y}^{MI}$ and the output of the slot modifier $\boldsymbol{y}^{MS}$, we apply the length normalization to keep the scale of reward stable and combine the normalized $\mathcal{L}_{MI}$ and $\mathcal{L}_{MS}$ to obtain the intent fit reward $R_{fit}^I$, the slot fit reward $R_{fit}^S$ and the final fit reward $R_{fit}$:

$$\mathcal{R}_{fit}^I = -\frac{1}{m}\mathcal{L}_{MI} \tag{9}$$

$$\mathcal{R}_{fit}^S = -\frac{1}{n}\mathcal{L}_{MS} \tag{10}$$

$$\mathcal{R}_{fit} = \alpha \mathcal{R}_{fit}^I + \beta \mathcal{R}_{fit}^S \tag{11}$$

where $m$ is the length of the output of the intent modifier $\boldsymbol{y}^{MI}$ and $n$ is the length of the output of the slot modifier $\boldsymbol{y}^{MS}$.

Secondly, the output of the modifier should not deviate too far from the reference, so we utilize a similarity reward $\mathcal{R}_{sim}$ to measure the similarity between the reference and the output of the modifier module. We use the accuracy of multiple intent detection and F1 score of slot filling as the similarity function to measure the similarity between $\boldsymbol{y_1}$ and $\boldsymbol{y_2}$, which are denoted as $\mathcal{S}_I(\boldsymbol{y_1}, \boldsymbol{y_2})$ and $\mathcal{S}_S(\boldsymbol{y_1}, \boldsymbol{y_2})$, respectively. Then the intent similarity reward $\mathcal{R}_{sim}^I$, the slot similarity reward $\mathcal{R}_{sim}^S$ and the final similarity reward $R_{sim}$ are formulated as:

$$\mathcal{R}_{sim}^I = \mathcal{S}_I(\hat{\boldsymbol{y}}^I, \boldsymbol{y}^{MI}) \tag{12}$$

$$\mathcal{R}_{sim}^S = \mathcal{S}_S(\hat{\boldsymbol{y}}^S, \boldsymbol{y}^{MS}) \tag{13}$$

$$\mathcal{R}_{sim} = \alpha \mathcal{R}_{sim}^I + \beta \mathcal{R}_{sim}^S \tag{14}$$

where $\hat{\boldsymbol{y}}^I$ is the reference intent and $\hat{\boldsymbol{y}}^S$ is and the reference slot. The suitability reward $\mathcal{R}_{suit}$ is the weighted sum of $\mathcal{R}_{fit}$ and $\mathcal{R}_{sim}$:

$$\mathcal{R}_{suit} = \lambda_f \mathcal{R}_{fit} + \lambda_s \mathcal{R}_{sim} \tag{15}$$

where $\lambda_f$ and $\lambda_s$ are two hyper-parameters.

Another common problem in many multi-intent SLU models is that the F1 score of slot filling and the accuracy of intent detection might increase or decrease asynchronously. As the F1 score of slot filling increases, the accuracy of intent detection might begin to decrease. Overall accuracy denotes the ratio of utterances whose intents and slots are all correctly predicted. As a result, it is crucial to achieve a trade-off between the accuracy of intent detection and the F1 score of slot filling.

Intuitively, we apply the overall accuracy to measure the similarity between $\boldsymbol{y_1}$ and $\boldsymbol{y_2}$, and denote it as $\mathcal{S}_A(\boldsymbol{y_1}, \boldsymbol{y_2})$. We directly apply $\mathcal{S}_A$ as the compromise reward $\mathcal{R}_{com}$ to improve the overall accuracy. The compromise reward $\mathcal{R}_{com}$ is:

$$\mathcal{R}_{com} = \mathcal{S}_A(\hat{\boldsymbol{y}}, \boldsymbol{y}^M) \tag{16}$$

The final reward $\mathcal{R}$ for the modifier module is:

$$\mathcal{R} = \mathcal{R}_{suit} + \lambda_c \mathcal{R}_{com} \tag{17}$$

| Model | MixATIS | | | MixSNIPS | | |
|---|---|---|---|---|---|---|
| | Overall(Acc) | Slot(F1) | Intent(Acc) | Overall(Acc) | Slot(F1) | Intent(Acc) |
| Attention BiRNN (Liu and Lane, 2016) | 39.1 | 86.4 | 74.6 | 59.5 | 89.4 | 95.4 |
| Slot-Gated (Goo et al., 2018) | 35.5 | 87.7 | 63.9 | 55.4 | 87.9 | 94.6 |
| Bi-Model (Wang et al., 2018b) | 34.4 | 83.9 | 70.3 | 63.4 | 90.7 | 95.6 |
| SF-ID (E et al., 2019) | 34.9 | 87.4 | 66.2 | 59.9 | 90.6 | 95.0 |
| Stack-Propagation (Qin et al., 2019) | 40.1 | 87.8 | 72.1 | 72.9 | 94.2 | 96.0 |
| AGIF (Qin et al., 2020) | 40.8 | 86.7 | 74.4 | 74.2 | 94.2 | 95.1 |
| LR-Transformer (Cheng et al., 2021b,a) | 43.3 | 88.0 | 76.1 | 74.9 | 94.4 | 96.6 |
| GISCo (Song et al., 2022) | 48.2 | 88.5 | 75.0 | 75.9 | 95.0 | 95.5 |
| SSRAN (Cheng et al., 2023a) | 48.9 | 89.4 | 77.9 | 77.5 | 95.8 | 98.4 |
| GL-GIN (Qin et al., 2021b) | 43.0 | 88.2 | 76.3 | 73.7 | 94.0 | 95.7 |
| w/ MRRL | 47.2$^\dagger$ | 88.7$^\dagger$ | 78.4$^\dagger$ | 75.8$^\dagger$ | 95.2$^\dagger$ | 96.5$^\dagger$ |
| Co-guiding Net (Xing and Tsang, 2022a) | 51.3 | 89.8 | 79.1 | 77.5 | 95.1 | 97.7 |
| w/ MRRL | 54.8$^\dagger$ | 90.6$^\dagger$ | 79.5$^\dagger$ | 79.0$^\dagger$ | 96.4$^\dagger$ | 98.6$^\dagger$ |
| ReLa-Net (Xing and Tsang, 2022b) | 52.2 | 90.1 | 78.5 | 76.1 | 94.7 | 97.6 |
| w/ MRRL | 55.8$^\dagger$ | 92.4$^\dagger$ | 79.8$^\dagger$ | 79.3$^\dagger$ | 96.8$^\dagger$ | 99.1$^\dagger$ |

Table 2: Results comparison. $\dagger$ denotes our model significantly outperforms baselines with $p < 0.01$ under t-test.

where $\lambda_c$ is a hyper-parameter. We utilize the RE-INFORCE algorithm (Williams, 1992; Zhang et al., 2021) to optimize the reward $\mathcal{R}$:

$$
\begin{aligned}
\nabla \mathcal{J} &= \nabla \sum_{\boldsymbol{y}^M} P_M(\boldsymbol{y}^{MI'}, \boldsymbol{y}^{MS} | \boldsymbol{x}, \hat{\boldsymbol{y}}) \mathcal{R} \\
&= \mathbb{E}_{\boldsymbol{y}^M \sim P_M} [\nabla \log P_M(\boldsymbol{y}^{MI'}, \boldsymbol{y}^{MS} | \boldsymbol{x}, \hat{\boldsymbol{y}}) \mathcal{R}]
\end{aligned}
\tag{18}
$$

## 4 Experiments

### 4.1 Datasets and Metrics

We conduct our experiments on two public multi-intent datasets[1]: cleaned version of MixATIS and MixSNIPS (Qin et al., 2020). MixATIS dataset is collected from ATIS dataset (Hemphill et al., 1990) and MixSNIPS dataset is collected from SNIPS dataset (Coucke et al., 2018). MixATIS includes 13,162 utterances for training, 756 utterances for validation and 828 utterances for testing. MixSNIPS includes 39,776 utterances for training, 2,198 utterances for validation and 2,199 utterances for testing. Compared to single-domain MixATIS dataset, MixSNIPS dataset is more complicated because of the intent diversity and large vocabulary.

Following Goo et al. (2018); Qin et al. (2021b), we evaluate accuracy (Acc) for multiple intent detection, F1 score for slot filling, and overall accuracy for the utterance-level semantic frame parsing. Overall accuracy denotes the ratio of the utterances whose intents and slots are all correctly predicted.

### 4.2 Implementation Details

We pre-train the model for 5K steps with a batch size 16 on each dataset. During both pre-training

[1] https://github.com/LooperXX/AGIF

and fine-tuning, we use Adam optimizer (Kingma and Ba, 2015) with $\beta_1 = 0.9, \beta_2 = 0.98$, and 4k warm-up updates to optimize parameters in our model. For all the experiments, we select the model which works the best on the dev set and then evaluate it on the test set. $\alpha$ is set to 0.9, $\beta$ is set to 0.1, $\lambda_f$ is set to 0.3, $\lambda_s$ is set to 0.2, and $\lambda_c$ is set to 0.5. All parameters are obtained by the annealing strategy (Ahn et al., 2019). Experiments are conducted at GeForce RTX 2080Ti and TITAN Xp.

### 4.3 Main Results

We introduce MRRL to many baselines. The results on the test sets is listed in Table 2, from which we have the following observations:

(1) Our MRRL consistently improves the performance of several baselines on all tasks and datasets. More encouragingly, our best variant (i.e. *ReLa-Net w/ MRRL*) achieves new state-of-the-art results. Specifically speaking, on MixATIS dataset, it overpasses the previous state-of-the-art model ReLa-Net by 3.6 and 2.3 on overall accuracy and slot filling, and overpasses the previous state-of-the-art model Co-guiding Net by 0.7 on multiple intent detection; on MixSNIPS dataset, it overpasses SS-RAN by 1.8, 1.0 and 0.7 on utterance-level semantic frame parsing, slot filling and multiple intent detection, respectively. This is because our methods provide a better training target for the non-autoregressive model by modifying the reference, which might be not be suitable for training due to the multi-modality problem.

(2) It is worth noting that the improvement on the MixATIS dataset is more obvious than that on

| Model | MixATIS | | | MixSNIPS | | |
|---|---|---|---|---|---|---|
| | Overall(Acc) | Slot(F1) | Intent(Acc) | Overall(Acc) | Slot(F1) | Intent(Acc) |
| ReLa-Net w/ MRRL | **55.8** | **92.4** | **79.8** | **79.3** | **96.8** | **99.1** |
| w/o $\mathcal{R}_{suit}$ | 53.6 ($\downarrow$1.8) | 91.3 ($\downarrow$1.1) | 78.8 ($\downarrow$1.0) | 78.1 ($\downarrow$1.2) | 95.3 ($\downarrow$1.5) | 98.2 ($\downarrow$0.9) |
| w/o $\mathcal{R}_{fit}$ | 54.4 ($\downarrow$1.4) | 91.6 ($\downarrow$0.8) | 79.2 ($\downarrow$0.6) | 78.4 ($\downarrow$0.9) | 95.6 ($\downarrow$1.2) | 98.5 ($\downarrow$0.6) |
| w/o $\mathcal{R}_{sim}$ | 54.2 ($\downarrow$1.6) | 91.8 ($\downarrow$0.6) | 79.4 ($\downarrow$0.4) | 78.6 ($\downarrow$0.7) | 95.8 ($\downarrow$1.0) | 98.7 ($\downarrow$0.4) |
| w/o $\mathcal{R}_{com}$ | 53.0 ($\downarrow$2.8) | 92.0 ($\downarrow$0.4) | 79.5 ($\downarrow$0.3) | 77.2 ($\downarrow$2.1) | 96.5 ($\downarrow$0.3) | 98.8 ($\downarrow$0.3) |
| w/o $\mathcal{R}_{suit}$ + More Parameters | 53.8 ($\downarrow$2.0) | 91.4 ($\downarrow$1.0) | 79.2 ($\downarrow$0.6) | 78.5 ($\downarrow$0.8) | 95.7 ($\downarrow$1.1) | 98.4 ($\downarrow$0.7) |
| w/o $\mathcal{R}_{com}$ + More Parameters | 53.4 ($\downarrow$2.4) | 92.2 ($\downarrow$0.2) | 79.6 ($\downarrow$0.2) | 78.8 ($\downarrow$0.5) | 96.6 ($\downarrow$0.2) | 98.9 ($\downarrow$0.2) |

Table 3: Results of ablation experiments of ReLa-Net on MixATIS dataset and MixSNIPS dataset.

the MixSNIPS dataset. We suspect that this is because MixSNIPS dataset is more complicated than MixATIS dataset. MixATIS dataset has a smaller vocabulary and fewer kinds of intentions and slots, where it is easier to propose a more suitable training target for non-autoregressive multi-intent models. As a result, the gain is greater on MixATIS dataset.

(3) The improvements in terms of overall accuracy are much sharper. This is because our method includes the compromise reward to overcome the misalignment problem. In this way, the correct predictions of the two tasks can be better aligned. As a result, more test samples get the correct utterance-level semantic frame parsing results, and then the overall accuracy is improved more significantly.

## 4.4 Analysis

We conduct a set of ablation experiments on ReLa-Net w/ MRRL to verify the advantages of MRRL from different perspectives, and the experimental results are shown in Table 3.

### 4.4.1 Effectiveness of Suitability Reward

Suitability reward is one of the key contributions of our MRRL, which is dedicated to solving the problem of unsuitability of the reference caused by the multi-modality problem. To verify this, we remove the suitability reward and refer it to *w/o $\mathcal{R}_{suit}$* in Tabel 3. We can clearly observe that overall accuracy drops by 1.8 on MixATIS and 1.2 on MixSNIPS, the slot F1 drops by 1.1 on MixATIS and 1.5 on MixSNIPS, and intent accuracy drops by 1.0 on MixATIS and 0.9 on MixSNIPS. When we only remove a component of $\mathcal{R}_{suit}$ (i.e. *w/o $\mathcal{R}_{fit}$* and *w/o $\mathcal{R}_{sim}$*), the performance also degrades in varying degrees. Following previous works (Qin et al., 2020, 2021b), to verify that the proposed suitability reward rather than the added parameters works, we increase the layers of intent decoder and slot decoder when $\mathcal{R}_{suit}$ is removed and refer it to *w/o $R_{suit}$ + More Parameters*. We could observe that despite the added parameters, it still performs

worse than *ReLa-Net w/ MRRL*, which suggests that the improvements come from the proposed suitability reward rather than involved parameters.

### 4.4.2 Effectiveness of Compromise Reward

To verify the effectiveness of compromise reward, we remove it and refer it to *w/o $\mathcal{R}_{com}$*. We can find that the performance is decreased on all tasks and datasets. Moreover, the drop in overall accuracy is more pronounced than that of slot F1 and intent accuracy on the two datasets, which suggests that compromise reward can efficiently improve the overall accuracy. It is worth noting that slot F1 and intent accuracy do not drop when compromise reward is introduced. We believe that the reason is that compromise reward can further achieve the mutual guidance between intent and slot indirectly when realizing the flexible trade-off between the two subtasks. Like Sec.4.4.1, we also increase the layers of intent decoder and slot decoder to verify that the compromise reward rather than the added parameters works, which is named as *w/o $\mathcal{R}_{com}$ + More Parameters*. The result also suggests that the improvements come from the compromise reward.

## 4.5 Case Study

To further demonstrate how our approach alleviates the multi-modality problem, we provide several cases generated from GL-GIN, ReLa-Net and ReLa-Net + MRRL in Figure 3.

It is obvious that despite the utilization of GAT, GL-GIN is still impacted by multi-modality issue, where B-airport_name is incorrectly predicted as B-fromloc.airport_name. We believe this is because GL-GIN has little prior knowledge about the reference during the inference progress, so *milwaukee* is incorrectly predicted as the departure place when there is no destination in the utterance. Compared to GL-GIN, ReLa-Net performs a little better, where the predicted intent is right but there are still some mistakes in the predicted slots. When MRRL is introduced to Rela-Net, the prediction is abso-

**Models**

| | | Utterance: | what | ground | transportation | is | available | between | milwaukee | airport | and |
|---|---|---|---|---|---|---|---|---|---|---|---|
| Ref. | Slot: | | O | O | O | O | O | O | B-airport_name | I-airport_name | O |
| | Intent: | atis_ground_service | | | | | | | | | |
| GL-GIN | Slot: | | O | O | O | O | O | O | B-fromloc.airport_name | I-fromloc.airport_name | O |
| | Intent: | atis_distance | | | | | | | | | |
| ReLa-Net | Slot: | | O | O | O | O | O | O | B-fromloc.airport_name | I-fromloc.airport_name | O |
| | Intent: | atis_ground_service | | | | | | | | | |
| ReLa-Net + MRRL | Slot: | | O | O | O | O | O | O | B-airport_name | I-airport_name | O |
| | Intent: | atis_ground_service | | | | | | | | | |

Figure 3: Cases that generated from GL-GIN (Qin et al., 2021b), ReLa-Net (Xing and Tsang, 2022b) and ReLa-Net w/ MRRL. The red text indicates the incorrect predictions.

lutely right, which indicates that MRRL can indeed alleviate the multi-modality problem.

## 5 Related Work

### 5.1 Intent Detection and Slot Filling

As deep learning obtains impressive performance on various tasks (Li et al., 2021, 2022; Zhang et al., 2022b; Yu et al., 2023; Zhang et al., 2023b; Li et al., 2023; Zhang et al., 2023a), more and more studies utilize deep learning to SLU and achieve notable achievements (Hakkani-Tür et al., 2016; Xia et al., 2018; Liu et al., 2019; Huang et al., 2020; Wu et al., 2020; Qin et al., 2021a,b; Huang et al., 2021, 2022; Chen et al., 2022a,b; Cheng et al., 2023e,b). Recently, the multi-intent SLU problem has garnered the significant attention, leading to the emergence of several graph-based models which have demonstrated promising results. AGIF (Qin et al., 2020) applies graph attention to directly connect the slot nodes of each token with all predicted intent nodes. GL-GIN (Qin et al., 2021b) further introduces a global-local graph interaction network specifically and leverages graph-based techniques to capture interactions between different parts of the input utterance. More recently, Xing and Tsang (2022a) proposes Co-guiding Net to enhance the overall performance via enabling slot and intent to guide and influence each other during the training process. Xing and Tsang (2022b) proposes ReLa-Net to further exploit label typologies and relations.

However, most of the previous models neglect the multi-modality problem and the misalignment problem, which are both detrimental to the perfor-

mance of the SLU model. Therefore, we introduce a modifier and propose a suitability reward to overcome the multi-modality problem and a compromise reward to overcome the misalignment problem and improve the overall accuracy.

### 5.2 Reinforcement Learning

Several NLP tasks have been solved through reinforcement learning techniques, such as dialogue generation (Li et al., 2016, 2017), question answering (Xiong et al., 2018; Lu et al., 2022), machine translation (Wu et al., 2018; Shao et al., 2023), sentiment transfer (Xu et al., 2018), and essay scoring (Wang et al., 2018c). In SLU task, Wang et al. (2018a) applies reinforcement learning to learn the wrong labeled slots with or without user's feedback, Rao et al. (2021) proposes a reinforce framework to enhance automatic speech recognition robustness in SLU. In our work, we apply reinforcement learning to alleviate the multi-modality problem and the misalignment problem in non-autoregressive SLU.

## 6 Conclusion

In this paper, we propose MRRL, a simple yet effective method to alleviate the multi-modality problem and misalignment problem in non-autoregressive multi-intent SLU. We introduce a modifier to provide a more suitable training target for the model, and apply reinforcement learning with the suitability reward and compromise reward. Experiments and analysis demonstrate the effectiveness of our proposed method, which can consistently improve the performance of baselines and the best variant achieves new state-of-the-art performance. Future

work will focus on how to further alleviate the two problems for non-autoregressive multi-intent SLU.

## Limitations

Although our MRRL consistently improve the performance of the baselines, and the best variant (i.e. ReLa-Net + MRRL) achieves new state-of-the-art results, it does not change the inherent structure of the model. In fact, the BiLSTM used is relatively simple, which limits the performance of SLU. In the future, we will pay more attention to these deficiencies and try to design better frameworks for non-autoregressive multi-intent SLU.

## Acknowledgements

We thank all anonymous reviewers for their constructive comments. This paper was partially supported by Shenzhen Science & Technology Research Program (No: GXWD20201231165807007-20200814115301001) and NSFC (No: 62176008).

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
