# OpenReview forum: "MRRL: Modifying the Reference via Reinforcement Learning for Non-Autoregressive Joint Multiple Intent Detection and Slot Filling"
_EMNLP/2023/Conference — EMNLP 2023 Findings_

### Official Review · Reviewer_ejsG · 2023-08-04

**Soundness:** 3

**Excitement:**

4: Strong: This paper deepens the understanding of some phenomenon or lowers the barriers to an existing research direction.

**Paper Topic And Main Contributions:**

This work explore a novel method to non-autoregressively solve multi-intent joint multiple intent detection and slot filling in a better manner. They find that existing non-autoregressive approaches still suffer from two issues: (1) the multi-modality problem that lead to reference intents and slots may not be suitable for training; (2) lack of alignment between the correct 008 predictions of the two tasks ,thereby limiting the performance. To solve the problem, this work introduce propose Modifying the Reference via Reinforcement Learning (MRRL) for intent detection and slot filling. Specifically, they provide a better training target for the non-autoregressive model by modifying the reference according to the output of the non-autoregressive model. Additionally, they also propose a compromise reward to realize a flexible trade-off between the two tasks.

**Reasons To Accept:**

(1)	The writing is clear, and experiments are extensive.
(2)	The introduced method can easily applied on many existing approaches.
(3)	The performance is good and significant (achieving stoa on two benchmarks).


**Reasons To Reject:**

(1) From the motivation, the intent modifier and slot modifier can provide a better training target, what is the different between the golden labels and modifier’s outputs? And why their output is the better target? The authors should provide more evidence.

(2)	The authors should provide more evidence to prove that their compromise reward can exactly help the alignment between the correct predictions of the two tasks. (like Figure 1)


**Reproducibility:**

3: Could reproduce the results with some difficulty. The settings of parameters are underspecified or subjectively determined; the training/evaluation data are not widely available.

**Reviewer Confidence:**

4: Quite sure. I tried to check the important points carefully. It's unlikely, though conceivable, that I missed something that should affect my ratings.

---

> ### Author Rebuttal · Authors · 2023-08-28
>
> Thank you for the insightful comments. The following are our responses to concerns raised:
>
> Q1: What is the different between the golden labels and modifier’s outputs? Why is their output the better target?
>
> A1: The golden labels are the ground truth and the modifier's outputs are the predictions. As we mentioned in $\textbf{line 319 - line 329}$, we design the fit reward to help the output of modifier fit well with the output of the non-autoregressive model and the similarity reward to help the output of the modifier module not deviate too far from the reference. Therefore, we believe the output of the modifier is the better target. Besides, we also conduct the ablation studies to verify the superiority of the modifier, as shown in Table 3. We will also add more evidence in the final version.
>
> Q2: More evidence to prove that their compromise reward can exactly help the alignment between the correct predictions of the two tasks.
>
> A2: Thanks for your advice. We will add more evidence in the final version to verify that the compromise reward could help the alignment between the correct predictions of the two tasks. We will show the learning curve like Figure 1 to help readers to better understand the training process.

---

### Official Review · Reviewer_cooG · 2023-08-05

**Soundness:** 4

**Excitement:**

3: Ambivalent: It has merits (e.g., it reports state-of-the-art results, the idea is nice), but there are key weaknesses (e.g., it describes incremental work), and it can significantly benefit from another round of revision. However, I won't object to accepting it if my co-reviewers champion it.

**Paper Topic And Main Contributions:**

This research applies reinforcement learning to address the multi-modality and misalignment problems in non-autoregressive multi-intent SLU. They introduce a new method called MRRL, which modifies the reference utterance for model training. MRRL shows promise, improving the performance of the baselines and achieving new state-of-the-art results.



**Reasons To Accept:**

- The paper identified two major challenges in non-autoregressive slot filling and intent detection. The authors proposed a "knowledge distillation" style approach to narrow the gap between model output and reference, which eases the training process. The result is great, boosting the performance of a series of baseline models.
- The architecture is easy to follow and the writing is clear.

**Reasons To Reject:**

- Further analysis is needed to demonstrate the necessicity of a modifier module. Why not use it as the final model? What if you use reinforcement learning directly on base models?
- More quantitative and qualitative analysis is needed to study the convergence and performance of RL-aided models. How does RL help balance the performance of two modules? For example, a similar learning curve like Figure 1 may help to better understand the training process.

**Reproducibility:**

4: Could mostly reproduce the results, but there may be some variation because of sample variance or minor variations in their interpretation of the protocol or method.

**Reviewer Confidence:**

4: Quite sure. I tried to check the important points carefully. It's unlikely, though conceivable, that I missed something that should affect my ratings.

---

> ### Author Rebuttal · Authors · 2023-08-28
>
> Thank you for your insightful comments. The following are our responses to concerns raised:
>
> Q1: Why not use the modifier as the final model?
>
> A1: Thanks for your question. Do you mean we only use the modifier as the final model? As we mentioned in $\textbf{line 294 to line 308}$. If we directly apply reinforcement learning to train the modifier, the modifier will be trapped in a non-optimal state where only sequences of meaningless intentions and slots are output, consisting of some of the most frequent intentions and slots, respectively. Therefore, we do not use the modifier as the final model. We first apply a pre-training strategy to establish a better initial state for the modifier so that the modifier can learn more easily in fine-tuning and avoid falling into local optimization. Then we fine-tune the modifier with two rewards in reinforcement learning.
>
> Q2: What if you use reinforcement learning directly on base models?
>
> A2: Thanks for your question. As shown in Figure 2, the base models and the modifiers share the parameters. As we mentioned in $\textbf{line 294 to line 302}$. If we directly apply reinforcement learning to train the modifier, the modifier will be trapped in a non-optimal state where only sequences of meaningless intentions and slots are output, consisting of some of the most frequent intentions and slots, respectively. In pre-training, we train the base models to establish a better initial state. And in fine-tuning, we fine-tune the modifier with two rewards in reinforcement learning. In the final version, we will further emphasize that they share the parameters.
>
> Q3: How does RL help balance the performance of two modules?
>
> A3: Thanks for your advice. We will add more quantitative and qualitative analysis in the final version to verify that RL could balance the performance of two modules. We will show the learning curve like Figure 1 to help readers to better understand the training process.

---

### Official Review · Reviewer_6FMz · 2023-08-05

**Soundness:** 4

**Excitement:**

3: Ambivalent: It has merits (e.g., it reports state-of-the-art results, the idea is nice), but there are key weaknesses (e.g., it describes incremental work), and it can significantly benefit from another round of revision. However, I won't object to accepting it if my co-reviewers champion it.

**Paper Topic And Main Contributions:**

This paper proposes a method for multi-intent detection and slot filling, which aims at tackling the two issues of non-autoregressive models: multi-modality and misalignment. The proposed method includes a modifier to modify the reference according to the output of non-autoregressive model, and a reward for applying reinforcement learning.
The proposed method is combined with several baselines, and the results show the improvement in comparison with the baselines.


**Questions For The Authors:**

1, Writing should be improved: you spend quite much content on related work and backgroud such as discussing the existing models in Section 2.
2, Table 2: the baselines, the proposed models, and the other models should be separated more clearly to follow easily


**Reasons To Accept:**

1, The proposed models show the improvement in comparison with the baselines.
2, The proposed method is useful to improve the tasks of dialogue models.
3, The proposed reward is quite interesting

**Reasons To Reject:**

1, More analyses should be conducted to clearly show how the issues (misalignment, and multi-modality) have been solved by the proposed method.


**Reproducibility:**

3: Could reproduce the results with some difficulty. The settings of parameters are underspecified or subjectively determined; the training/evaluation data are not widely available.

**Reviewer Confidence:**

4: Quite sure. I tried to check the important points carefully. It's unlikely, though conceivable, that I missed something that should affect my ratings.

---

> ### Author Rebuttal · Authors · 2023-08-28
>
> Thank you for the insightful comments. The following are our responses to concerns raised:
>
> Q1: More analyses should be conducted to clearly show how the issues (misalignment, and multi-modality) have been solved by the proposed method.
>
> A1: Thanks for your advice. We will add more analyses and provide more examples in the final version to show how the issues (misalignment, and multi-modality) have been solved by the proposed method.
>
> Q2: Quite much content on related work and background.
>
> A2: Thanks for your advice. We introduce the details of related work and background to help the readers  better understand the task. We take your advice and will move some of the details of Section 2 to the appendix in the final version.
>
> Q3: Table 2: the baselines, the proposed models, and the other models should be separated more clearly to follow easily.
>
> A3: Thanks for your advice. We will separate them more clearly to follow easily in the final version.

---

### Meta-Review · Area_Chair_YgTF · 2023-09-17

**Recommendation:** 3

**Metareview:**

This work applies reinforcement learning (RL) to address the multi-modality and misalignment problems in non-autoregressive multi-intent SLU. It introduces a new method called Modifying the Reference via Reinforcement Learning (MRRL).

The paper identifies two interesting challenges in non-autoregressive slot filling and intent detection, demonstrates good performance and proposes an interesting reward. However, the reviewers considered that additional analysis would be helpful in determining how the issues are being addressed, and the stability of the proposed RL approach.

---

### Decision · Program_Chairs · 2023-10-07

**Decision:**

Accept-Findings

**Comment:**

This work applies reinforcement learning (RL) to address the multi-modality and misalignment problems in non-autoregressive multi-intent SLU. It introduces a new method called Modifying the Reference via Reinforcement Learning (MRRL).

The paper identifies two interesting challenges in non-autoregressive slot filling and intent detection, demonstrates good performance and proposes an interesting reward. However, the reviewers considered that additional analysis would be helpful in determining how the issues are being addressed, and the stability of the proposed RL approach.